# Dynamic Soaring Parameters Influence Regularity Analysis on UAV and Soaring Strategy Design

**Wei Wang** , **Weigang An * and Bifeng Song**

School of Aeronautics, Northwestern Polytechnical University, Xi'an 710072, China;
vulpes.wang@mail.nwpu.edu.cn (W.W.)
* Correspondence: anweigang@nwpu.edu.cn; Tel.: +86-188-2178-5826

**Abstract:** Dynamic soaring helps albatross achieve long-distance migration. From a bionic view, dynamic soaring has great potential to enhance unmanned aerial vehicles ("UAVs") flight range and endurance. The previous application studies focus on flight strategies to guide UAV soaring. However, the energy harvesting efficiency problem emerges. The lack of clear dynamic soaring influencing factors hinders dynamic soaring UAV design and flight strategy design from a theoretical perspective. Hence this paper aims to analyze the influence law of different UAV mass, initial airspeed, and entering angle. Trajectories and flight data in different factors are obtained through trajectory optimization. The results show that UAV mass has a positive influence on energy harvesting. The initial airspeed and entering angle affect both energy efficiency and trajectory. For UAV design, weight balance needs to be considered rather than a pursuit of the lightest. For flight strategy design, finding an optimal initial state will improve energy efficiency.

**Keywords:** dynamic soaring; influence regulation; bionic UAV design

## 1. Introduction

Many seabirds, such as the albatross, can use horizontal winds to travel long distances without flapping wings. Their unique flight mode attracted the attention of scientists while they glided freely over the ocean. In the late 19th century, Lord Rayleigh first named this flight mode as dynamic soaring [1]. Scientists monitored the flight of the albatross and found that it can fly continually over 13 days [2], satellites tracked albatross' flight trajectory and the images are shown in Figure 1. The ability to harvest energy from wind gives dynamic soaring great potential in the bionic long-endurance flight. If dynamic soaring is applied to unmanned aerial vehicles (UAVs) flight, it has great potential to improve flight distance and endurance, which are the core performance of UAVs.

Although scientists have studied dynamic soaring from different angles, the application of UAVs dynamic soaring is still hard to achieve. One of the challenges is the efficiency of energy harvesting. Current research conclusions and simulation results show that the UAVs' parameters significantly influence the dynamic soaring performance. These lead to questions about parameters influence law and how to design a well-performance dynamic soaring UAV. Hence, this paper focuses on the influence of UAVs' parameters and gives a regularity to help dynamic soaring UAV design and flight strategy design.

Since Lord Rayleigh first defined albatross flight mode as dynamic soaring, many studies have been carried out in the past decades. Scientists summarized albatross' basic background and flight modes [3,4]. For the albatross' flight mechanism, in 1925 Walkden first proposed to use the dynamics model of aircraft for dynamic soaring force analysis and mechanism derivation [5]. In 2019 and 2020, Sachs fully reviewed the dynamic soaring mechanism study and presented a complete mechanism model [6,7]. Sachs solved a conflict about reference frame choice on the dynamic soaring mechanism model built and answered how albatross fly. For the albatross' aerodynamic characteristics, in 2018,

Stempeck designed a UAV with albatross wing shape [8]. The results showed that the wing shape inspired by albatrosses had a better lift-to-drag ratio at low angles of attack. Rice investigated the color of albatross wings and applied it to a UAV [9]. The total drag was reduced by 12.8% compared to a regular drone wing.

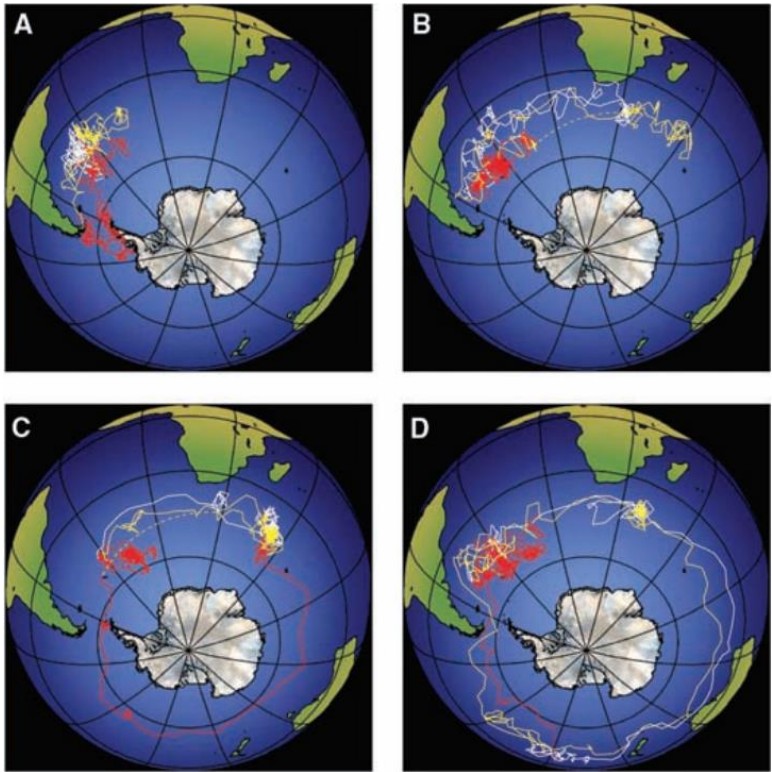

**Figure 1.** GPS tracking for albatross long endurance flight [2]. (**A–D**) represent albatross different soaring trajectories.

To achieve dynamic soaring by UAVs, researchers started to study dynamic soaring flight strategies. In the past decades, the trajectory optimization method was explored. Trajectory optimization, which involves producing flight paths that closely approximate real-world dynamics, can be thought of as a form of flight simulation. Sachs has significantly contributed by using this method to solve many trajectories, which can guide UAVs to realize dynamic soaring flight [10–15]. A very typical use of dynamic soaring flight strategy is for radio-controlled (RC) UAVs. These UAVs can use wind shear near ridges to accelerate up to 243.6 m/s. Sachs enhanced this flight mode with a more optimal O-shape trajectory [10], resulting in a maximum speed of 268.2 m/s. The trajectory optimization study presented many trajectories which guaranteed UAV flight without energy consumption. However, these trajectories were not generated in real time.

Recently, a more adaptive online trajectory optimization was proposed and studied. In 2009, Hong proposed a coupling method for offline and online trajectory optimization based on real-time parameters [16,17]. Hong linked the parameters with the wind field and updated the optimization result online. However, the trajectory optimization method needs high calculation resources and a long solution time, which are key challenges for UAVs' applications. Scientists have tried to find other flight strategy methods to guide UAVs to achieve dynamic soaring. In 2009, Lawrance designed a piecewise trajectory dynamic soaring control to solve the high calculation time problem [18]. Lawrance's method's basic idea is to link the flight parameters with energy change and give the UAV control command by the flight parameters. In 2022, Wang proposed a segmented dynamic soaring strategy [19]. Meanwhile, reinforcement learning also is used to achieve dynamic soaring. In 2020, Li proposed a heuristic control method to solve the dynamic soaring strategy

problem [20]. However, with well-developed flight strategies, UAVs still perform poor energy-gaining efficiency.

To solve the energy harvesting efficiency problem, scientists compared albatross flights and UAV flights. Parameters' influence draws researchers' attention. In 2015, Shan explored the influence of the flight Euler angles and wing load [21]. In 2018, Mir explored the energy harvesting performance with wingspan change [22,23]. Mir noticed that the albatross had significant wing deformation during the dynamic soaring. After wingspan changes, the minimum wind gradient was reduced by 15%, and the maximum speed increased by 6.2%. The average energy harvesting rises by 9%. Sachs also found the influence of sweep angle in high-speed dynamic soaring [24]. However, these studies only determined that partial parameters influence dynamic soaring performance but remain unknown variations.

From previous studies, UAV dynamic soaring application faces energy efficiency issues. The fundamental problem is the unclear influence factors and regularity. The unknown parameters effect make researchers hard to select or design dynamic soaring UAV and chose appropriate flight strategies to guide UAV flight. Hence, the uncertainty of influence factors leads to unsatisfactory energy harvesting efficiency. This paper explores the influence between flight parameters and UAV dynamic soaring performance. From the theoretical side, exploring the parameters' influence can help understand their detailed impact during the dynamic soaring. From the applications side, it will help to design dynamic soaring UAVs and improve the flight strategy. Finally, this makes UAVs' dynamic soaring flight come true.

This paper selects UAV mass, initial airspeed, and entering angle (the angle between wind direction and airspeed vector at the beginning) as influence factors from the dynamic soaring mechanism model. This paper first uses the two-step dynamic soaring trajectory optimization to solve the minimum required wind gradient and then the maximum energy gain. The trajectory and flight parameters variations are obtained. Then this paper analyzes these results from the energy harvesting, trajectory shape, and detailed phases. The results show that mass has a positive effect on energy harvesting. The heavier the UAV, the more energy it gains. Initial speed and entering angle significantly influence the energy-gaining and flight strategy. The optimal initial state emerges with the best dynamic soaring performance.

The basic models of dynamic soaring mechanism, UAV, and trajectory optimization are built in Section 2. From Section 2, the influence factors can be selected. Section 3 summarizes the result of different influence factors on dynamic soaring. Moreover, Section 3 discusses how to improve dynamic soaring UAV design and flight strategy. Section 4 presents the conclusions about this paper.

## 2. Parameters Selection and Models Establishment

In order to explore the influence rule of UAVs' parameter changes on dynamic soaring performance, three major steps are needed. The dynamic soaring energy harvesting mechanism model needs to be built to extract influencing factors. Then the UAV model and flight mission are determined. Finally, the optimal problem of the maximum energy harvesting of the same flight mission needs to be formulated. The mechanism model and flight mission are established through mathematical deduction. The optimization problem is formulated using deduction and solved using GPOPS-II.

### 2.1. Parameters from the Mechanism Model

The dynamic soaring energy harvesting mechanism model indicates the energy changing rate through the energy harvesting equation and answers where the energy comes from. The parameters involved in this model may be the influencing factors. This model is based on the force, motion, and mechanical energy equation. Therefore, this paper uses mathematical deduction to obtain the energy harvesting mechanism model and selects influencing factors from it.

From flight dynamics, the UAV force model can be illustrated in Figure 2:

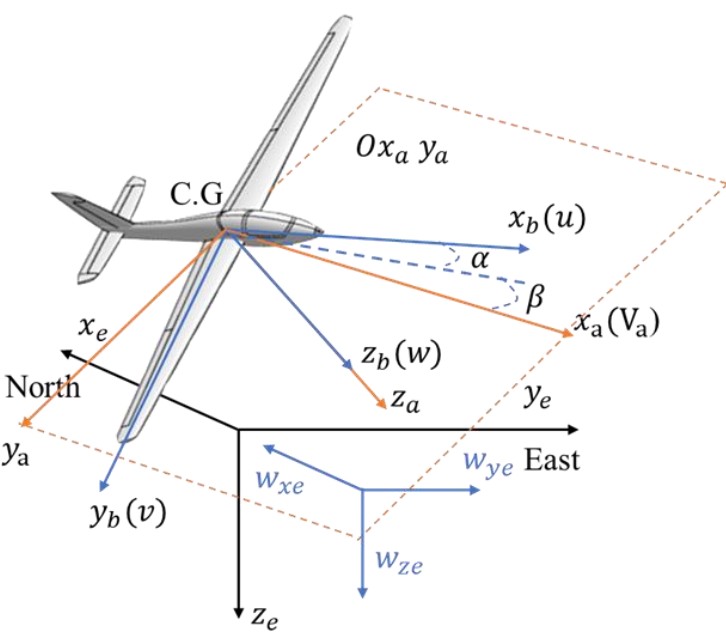

**Figure 2.** UAV force model in the three coordinate systems (North-East-Down axes, Body axes, Wind axes).

The airspeed of UAV and wind speed can be decomposed into three directions in Equations (1) and (2).

$$\overrightarrow{V_{uav}} = u\overrightarrow{i_b} + v\overrightarrow{j_b} + w\overrightarrow{k_b},\tag{1}$$

$$\overrightarrow{W} = W_x\overrightarrow{i_e} + W_y\overrightarrow{j_e} + W_z\overrightarrow{k_e},\tag{2}$$

In the non-inertial reference frame, the mechanical energy of a UAV can be expressed in Equation (3):

$$E = \frac{1}{2}m\left(u^2 + v^2 + w^2\right) - mgz_e,\tag{3}$$

$z_e$ is the vertical axis in the North-East-Down axes (NED axes), the minus is used for correct potential energy due to the NED axes defining height in the negative axis.

Derivative of the mechanical energy equation, the change rate of energy can be obtained in Equation (4):

$$\frac{dE}{dt} = m\frac{du}{dt}u + m\frac{dv}{dt}v + m\frac{dw}{dt}w - mgV_{ze},\tag{4}$$

where $V_{ze}$ is the ground speed in the vertical direction in the NED axes. Meanwhile, the equation of motion of the UAV is as follows in Equation (5):

$$m\begin{bmatrix} \frac{du}{dt} \\ \frac{dv}{dt} \\ \frac{dw}{dt} \end{bmatrix} = T_N^B\begin{bmatrix} F_{xe} \\ F_{ye} \\ F_{ze} \end{bmatrix} + T_W^B\begin{bmatrix} F_{xa} \\ F_{ya} \\ F_{za} \end{bmatrix} - m\begin{bmatrix} qw - rv \\ ru - pw \\ pv - qu \end{bmatrix} - T_N^B\begin{bmatrix} m\dot{W}_{xe} \\ m\dot{W}_{ye} \\ m\dot{W}_{ze} \end{bmatrix},\tag{5}$$

where $T_N^B$ is the transformation matrix from the NED axes to the body axes, $T_W^B$ is the transformation matrix from the wind axes to the body axes, $F_e$ is the force defined in the NED axes such as gravity, $F_a$ is the force defined in the wind axes such as aerodynamic forces, $[p\ q\ r]$ is the angular velocity in three directions in the body axes, $\dot{W}_e$ is the wind acceleration.

Substitute Equation (5) into the kinetic energy part of Equation (4) to obtain the change rate of kinetic energy in a non-inertial reference frame. The parameters in the new

equation from the body coordinate system can be converted to the ground coordinate and airflow coordinate systems. Finally new equation can be simplified and combined with the potential energy change rate as follows in Equation (6):

$$\frac{dE}{dt} = F_{xe}(V_{xe} - W_{xe}) + F_{ye}(V_{ye} - W_{ye}) + F_{ze}(V_{ze} - W_{ze}) + F_{xa}V_a - mW_{xe}(V_{xe} - W_{xe}) - m\dot{W}_{ye}(V_{ye} - W_{ye}) - m\dot{W}_{ze}(V_{ze} - W_{ze}) - mgV_{ze},$$ (6)

Convert wind acceleration to horizontal wind gradient in Equation (7):

$$m \begin{bmatrix} \dot{W}_{xe} \\ \dot{W}_{ye} \\ \dot{W}_{ze} \end{bmatrix} = \begin{bmatrix} \frac{dW_{xe}}{dt} \\ \frac{dW_{ye}}{dt} \\ \frac{dW_{ze}}{dt} \end{bmatrix} = \begin{bmatrix} \frac{dW_{xe}}{dz}\frac{dz}{dt} \\ \frac{dW_{ye}}{dz}\frac{dz}{dt} \\ \frac{dW_{ze}}{dz}\frac{dz}{dt} \end{bmatrix} = \begin{bmatrix} -G_x V_{ze} \\ -G_y V_{ze} \\ -G_z V_{ze} \end{bmatrix},$$ (7)

In general, the force defined in the NED axes is gravity, and the force defined in the wind axes is aerodynamic force. By substituting the above gradients and forces, the energy harvesting equation can be obtained as follows in Equation (8):

$$\frac{dE}{dt} = mg(-W_{ze}) - DV_a + mG_x V_{ze}(V_{xe} - W_{xe}) + mG_y V_{ze}(V_{ye} - W_{ye}) + mG_z V_{ze}(V_{ze} - W_{ze}),$$ (8)

The energy harvesting mechanism model reflects the energy change rate during the UAV dynamic soaring. The parameters in the equation that may affect energy harvesting are referred to as influencing factors. Mass, speed, wind gradient, and drag intuitively impact energy harvesting. Further, the model includes ground speed and airspeed. Hence the aerodynamic angle between different speeds may influence the energy harvesting.

From the UAVs' side, mass is an essential parameter in UAV design. Meanwhile, the mass will change during the flight due to fuel consumption or load delivery. The UAV mass variations influence during the dynamic soaring is a key problem for soaring performance. Further, from the flight strategy point of view, different initial flight states will influence soaring performance. Hence finding out what state is best for dynamic soaring will help to build a better soaring strategy. Based on the above views, this paper selects UAV mass, initial airspeed, and initial entering angle as influence factors to carry out research.

### 2.2. UAV and Flight Model Establishment

This study needs the specific UAV model, wind field model, and flight mission. This section will show the above model in detail. The Fox glider is chosen as the dynamic soaring UAV model. Its basic information is as follows in Figure 3 and Table 1.

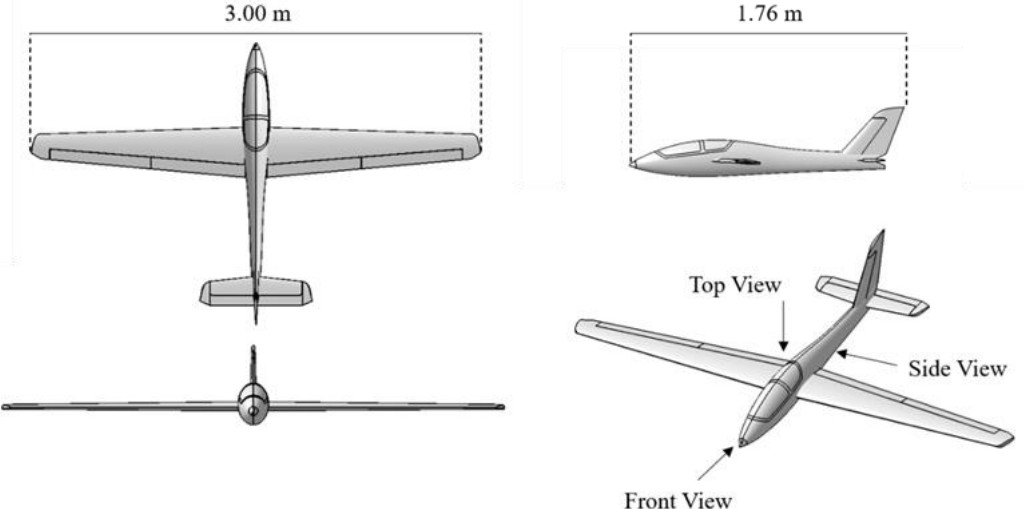

**Figure 3.** Three views of the glider Fox.

**Table 1.** Parameters of the glider Fox.

| Parameters, Unit | Value |
|---|---|
| Mass, kg | 4.7 |
| Maximum Takeoff Weight, kg | 5.1 |
| Wingspan, m | 3 |
| Wing Area, m$^2$ | 0.75 |
| Aspect Radio | 11.57 |
| $C_{Lmax}$ | 1.5 |
| $C_{D0}$ | 0.0223 |
| $L/D_{max}$ | 27.96 |

For the wind field selection, the current standard wind field models include the linear wind field model, exponential wind field model, and logarithmic wind field model. This paper chooses the linear wind field model to ensure the control variable because the wind gradient is constant. During the exploration of influencing factors, parameter changes may lead to flight trajectory changes, which cause wind gradient changes in the nonlinear wind field model. Hence the linear model has an advantage in the variable study. This paper chooses a linear wind field from west to east without wind speed on the ground.

The typical dynamic soaring cycles for the flight mission selection are open-loop and closed-loop trajectories. This study considers the application of UAVs and chooses the closed-loop as the research trajectory. The flight mission in the closed-loop flight can be described as starting at the initial point and flying around unpowered and back to the starting point. The closed-loop flight mission ensures that the potential energy is unchanged after the flight. Once requested, the airspeed is the same as the start; the kinetic energy is also unchanged. This condition is called energy-neutral. Specific parameters for the flight mission are shown in Table 2.

**Table 2.** Parameters of the flight mission.

| Initial Point, Unit | Final Point, Unit |
|---|---|
| $[x_0, y_0, z_0] = [0, 0, 10]$ (m) | $\left[x_f, y_f, z_f\right] = [0, 0, 10]$ |
| $V_0 = 20$ (deg) | $V_f = 20/\text{Unlimited}$ (deg) |
| $\gamma_0 = 0$ (deg) | $\gamma_f = \text{Unlimited}$ (deg) |
| $\psi_0 = 0$ (deg) | $\psi_f = \text{Unlimited}$ (deg) |

The $[x_0, y_0, z_0]$ and $\left[x_f, y_f, z_f\right]$ are the coordinates of UAV in the NED axes, $V$ is the airspeed, $\gamma$ is the climb angle, and $\psi$ is the yaw angle. The research model in this paper can be expressed in Figure 4:

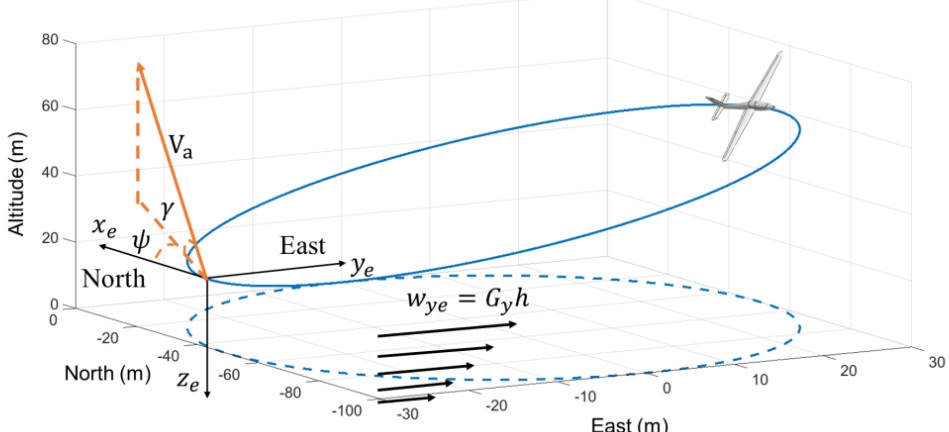

**Figure 4.** Flight mission and wind field model in this paper.

The energy harvesting mechanism model can be simplified as Equation (9):

$$\frac{dE}{dt} = -DV_a + mG_y V_a^2 sin\gamma cos\gamma sin\psi, \tag{9}$$

To clarify the aerodynamic forces applied in the equation, the first term will be further deduced. The drag can be calculated by the drag coefficient, airspeed, and wing area, shown in Equation (10):

$$\frac{dE}{dt} = -\frac{1}{2}\rho S(C_{D0} + KC_L^2)V_a^3 + mG_y V_a^2 sin\gamma cos\gamma sin\psi, \tag{10}$$

where $\rho$ is air density. $S$ is the wing area. $C_{D0}$ is the zero-lift drag coefficient. $K$ is the induced drag factor. $C_L$ is the lift coefficient.

### 2.3. Optimal Control Problem Establishment

This paper uses optimal control as the numerical method to solve the optimization problem, specifically, by using the direct collocation method to transform the dynamic soaring trajectory optimization problem into nonlinear programming, and choosing the solver based on the Gaussian Pseudo-spectral method to calculate the UAV dynamic soaring trajectory and obtain the maximum energy gained. The optimal control problem of this study can be expressed as follows.

To determine the minimum wind gradient for the dynamic soaring by the glider Fox, the objective is to minimize the wind gradient, shown in Equation (11).

$$min\ J = G_y, \tag{11}$$

To explore the maximum energy harvesting, the objective is to maximize the cost function as Equation (12).

$$max\ J = \int_o^{t_f} \left(mV_a\dot{V}_a - mg\dot{z}_e\right), \tag{12}$$

The system dynamic can be written as Equations (13)–(18):

$$\dot{V}_a = -D/m - gsin\gamma - \dot{W}_{ye}sin\psi cos\gamma, \tag{13}$$

$$\dot{\psi} = \frac{\left(Lsin\phi + -m\dot{W}_{ye}cos\psi\right)}{mV_a cos\gamma} \tag{14}$$

$$\dot{\gamma} = \frac{\left(Lcos\phi - mgcos\gamma + m\dot{W}_{ye}sin\psi sin\gamma\right)}{mV_a} \tag{15}$$

$$\dot{x}(North) = V_{xe} = V_a cos\gamma cos\psi \tag{16}$$

$$\dot{y}(East) = V_{ye} = V_a cos\gamma sin\psi + W_{ye} \tag{17}$$

$$\dot{z} = V_{ze} = V_a sin\gamma \tag{18}$$

In this study, the state variables are $[x, y, z, V_a, \gamma, \psi]$, the control variables are $[\phi, C_L]$. All variables are linked by system dynamic. Further, the optimal control problem needs constraints and boundaries. For dynamic soaring problems, sharp turns are involved. Hence the path constraint is the overload. The boundaries are related to the flight mission, and Table 2 illustrates these boundaries.

To solve the dynamic soaring optimal control problem, this paper uses GPOPS-II as the numerical solver. Specifically, the nonlinear programming (NLP) solver is SNOPT, and the optimization method used is "hp-PattersonRao", with a mesh tolerance of $1 \times 10^{-5}$. The calculation time is around 2.5 s.

## 3. Results and Discussion

This section has studied the influence of mass, initial airspeed, and entering angle on the UAV dynamic soaring energy harvesting. The optimization objective is the maximum energy gain. The influencing factors under the same wind field and flight mission are explored, and the reasons and rules of the influence are further analyzed, providing solutions for applying dynamic soaring.

### 3.1. Determination of Minimum Wind Gradient

Before the research of regularity, the minimum wind gradient for the glider Fox to achieve dynamic soaring in the set flight mission is required. After determining the minimum wind gradient, an environmental wind gradient larger than the minimum is set to ensure that the UAV can obtain energy.

By limiting the terminal airspeed and position, this paper ensures that the kinetic energy and potential energy are unchanged. This is the energy-neutral mentioned above, and the wind gradient for this situation is minimum. After calculating by using Gaussian Pseudospectral, the minimum wind gradient required by the UAV for dynamic soaring is 0.1536/s, and the energy-neutral trajectory is obtained in Figure 5.

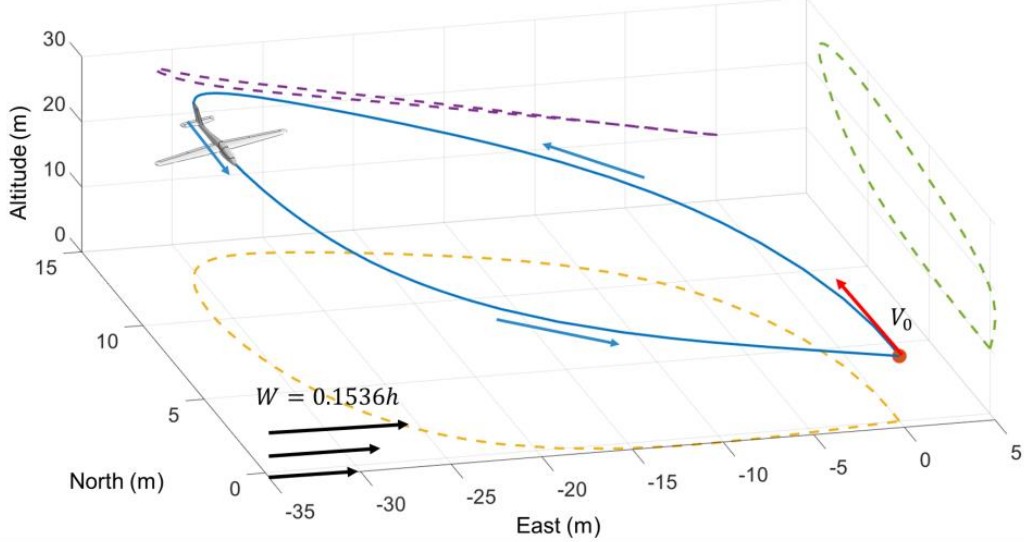

**Figure 5.** The trajectory of a closed-loop flight mission under the minimum wind gradient.

It can be found that the UAV starts from the initial point and uses dynamic soaring to achieve the flight mission of returning to the start without power. According to the theory of dynamic soaring, a typical cycle can be divided into four phases: Windward climb, high altitude turn, leeward descent, and lower altitude turn. This paper uses the yaw angle to divide the four phases in Figure 6.

By observing the energy variation curves and the division, it can be found that the four phases based on the yaw angle are consistent with the energy change trend, which can verify the feasibility of using the yaw angle to divide the dynamic soaring phases. Moreover, it is found that energy harvesting is mainly in the windward climb and leeward descent, and the energy consumption mainly occurs in the turning stage.

Once determined the minimum wind gradient required for the UAV to complete dynamic soaring in the flight mission in this paper, the wind gradient could be increased to maximize the energy harvesting, and the influencing factors could be studied. The wind

gradient for the influencing factors research is raised to 0.18/s to ensure that the UAV can obtain energy under the current flight mission.

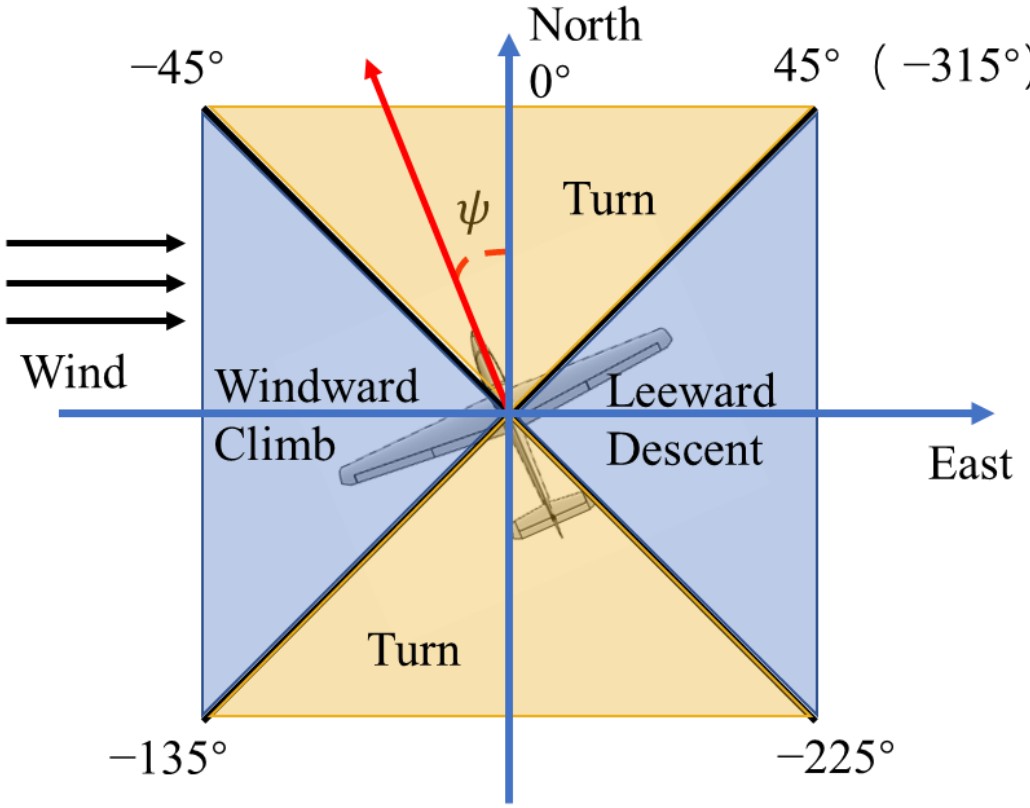

**Figure 6.** The four phases are determined by the yaw angle.

Based on the above phases' division, the energy variation curves are drawn in Figure 7.

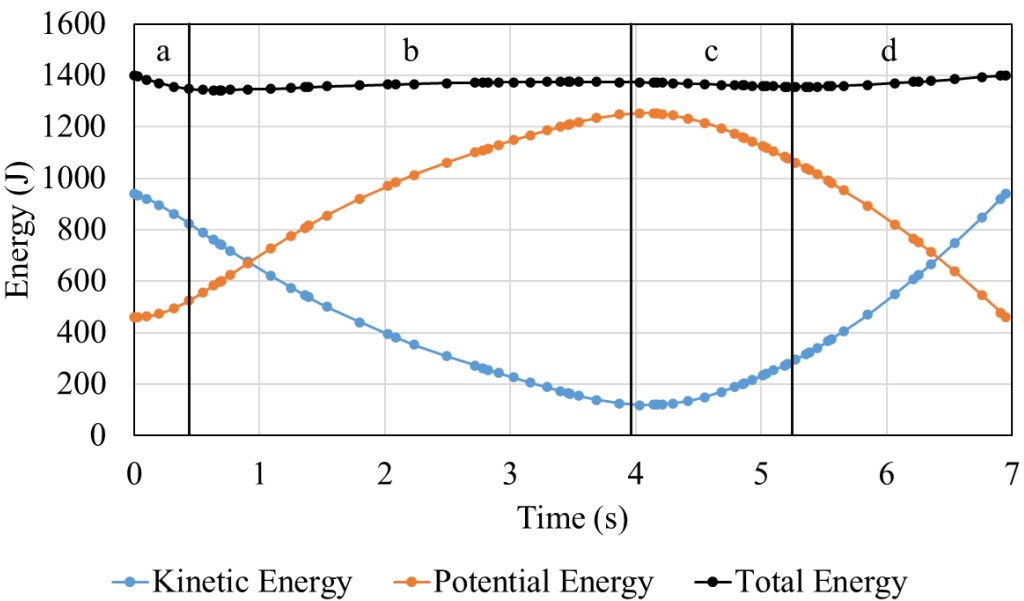

**Figure 7.** The energy variation curves in three types of energy. (**a**) Lower altitude turn, (**b**) Windward climb, (**c**) High altitude turn, (**d**) Leeward Descent.

### 3.2. Influence of Mass on Maximum Energy Harvesting

The weight of the glider Fox usually is 4.7 kg, and the maximum weight is 5.1 kg with a full payload. This section explores the influence of mass on maximum energy harvesting within the range that the UAV can take. Starting at 4.7 kg and increasing to 5.1 kg at 100 g intervals, the energy changes of the same flight mission with different masses are obtained as shown in Figure 8 and Table 3.

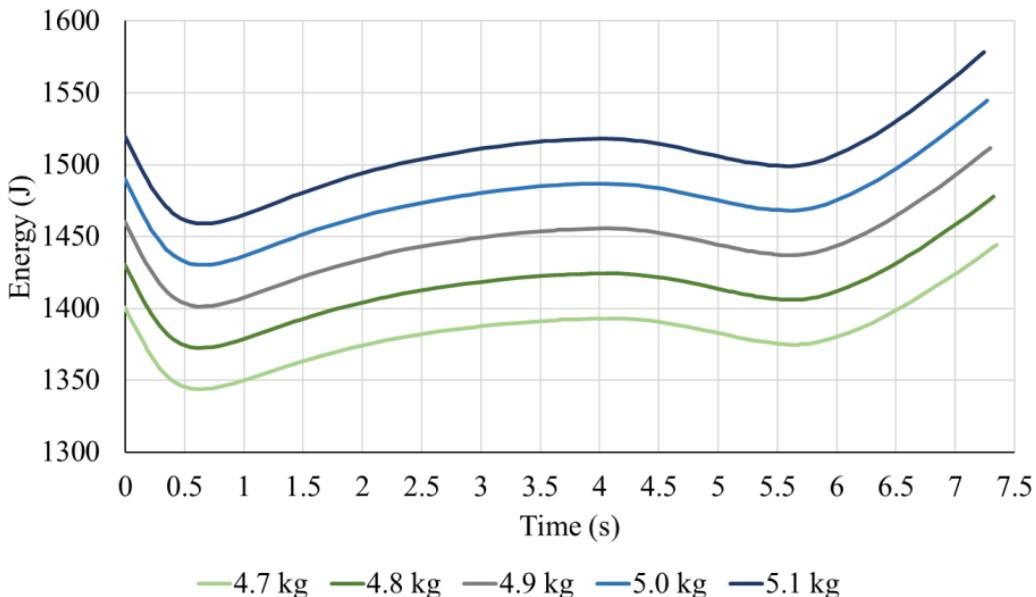

**Figure 8.** The energy variation curves in different masses as a function of time.

**Table 3.** Total energy changes in different masses.

| Mass, kg | Total Energy Changes, J |
|---|---|
| 4.7 | 43.51 |
| 4.8 | 47.42 |
| 4.9 | 51.25 |
| 5.0 | 55.02 |
| 5.1 | 58.70 |

From Figure 8 and Table 3, it can be found that the energy gained under the same flight mission increases with the increase in UAV's weight. Comparing different masses, the trend of energy change is the same. The obvious change is the initial and final value of the energy.

After comparison, the energy increases linearly and can be obtained by function fitting in Equation (19):

$$\Delta E = 37.98m - 134.92, \tag{19}$$

The reason for the linear energy change can be found in the energy harvesting mechanism. From the energy harvesting mechanism, the energy gaining can be expressed as the integral of the energy harvesting equation, such as Equation (20).

$$\Delta E = \int_0^{t_f} (-DV_a)dt + m \int_0^{t_f} (G_y V_a^2 sin\gamma cos\gamma sin\psi)dt, \tag{20}$$

Notice that mass exists as a first term in the energy harvesting term, so the energy gaining increases linearly with the increase in mass. The result is consistent with the analysis of the mechanism model.

For the energy harvesting trajectory, the different masses are plotted in Figure 9:

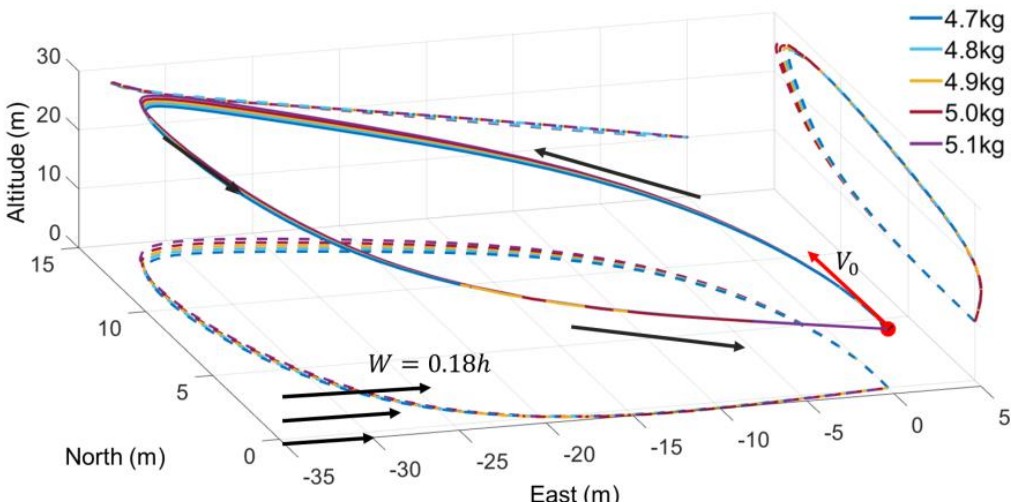

**Figure 9.** Trajectories for different masses.

By observing the dynamic soaring trajectories under different masses, it can be found that the trajectories do not change significantly with the mass increase. The windward climb distance in the 5.1 kg UAV is slightly increased by 0.98% compared with the 4.7 kg UAV, and the lower altitude turn distance is increased by 8.56%. Hence, from the perspective of the dynamic soaring application, under the same conditions, the UAV payload change will not significantly impact the determined dynamic soaring trajectory.

The energy changes in the four dynamic soaring phases are observed based on the trajectory and overall energy harvesting trend. Consistent with the previous exploration results, the energy gains in the windward climb and leeward descent phases, while the energy consumes in the turning phases.

The energy comparison diagram (Figure 10) shows that with the increase in mass, the values of both energy gain and loss increase. However, compared with the energy gain, energy consumption increases slowly. The difference between energy gain and loss is total energy gain, which also continues to increase. The energy gained by the UAV in the 5.1 kg increases by 17.01% compared with 4.7 kg. The energy lost 5.44% in the turning phases. The increment difference between the two is one order of magnitude, so the total energy increases. Take a closer look at the time ratio of each stage in Figure 11.

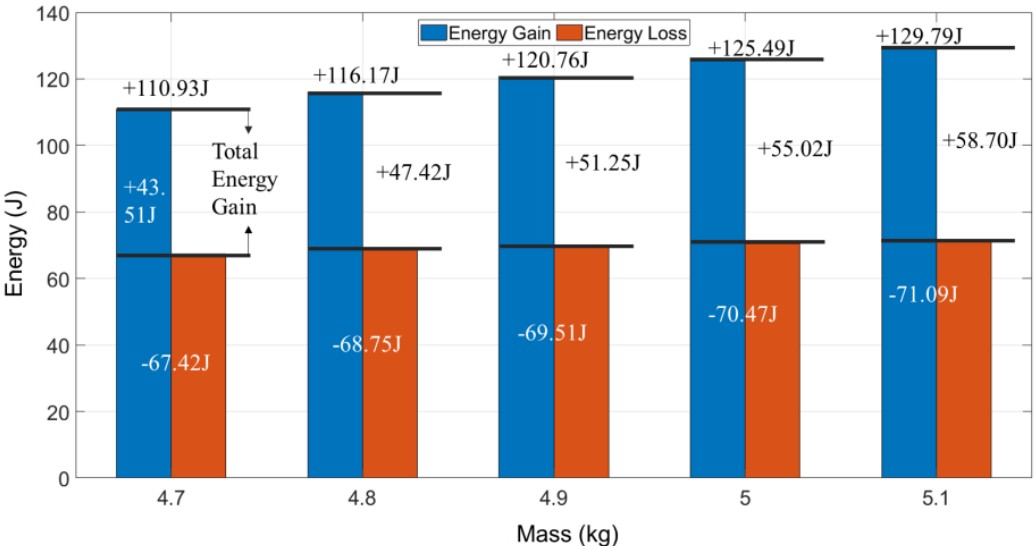

**Figure 10.** Comparison of energy gain and loss in different masses.

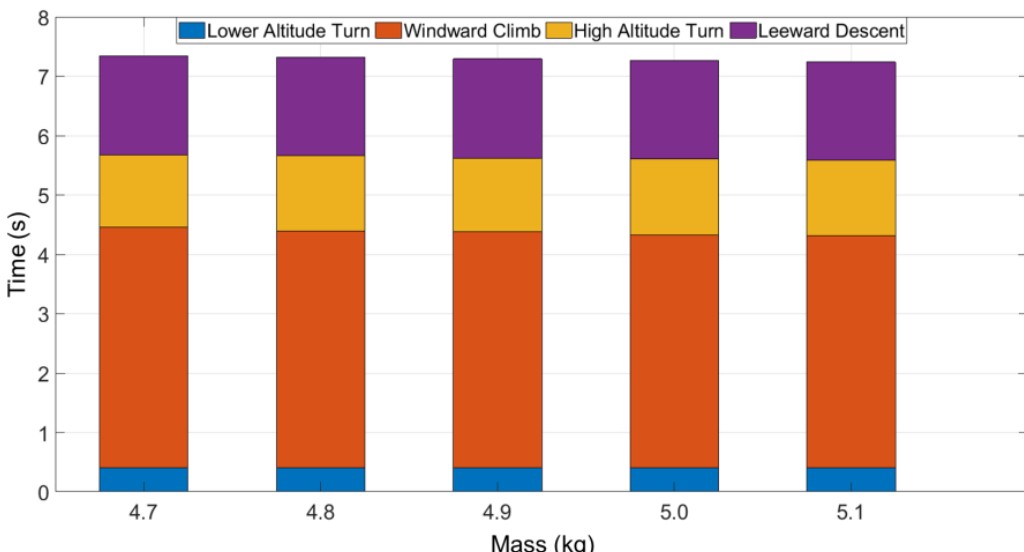

**Figure 11.** Time distribution in different masses.

During the energy harvesting period, the windward climb takes the longest time, but less energy gaining, and the leeward descent takes the shortest time but more energy gaining. With the increase in mass, this law does not change significantly. The average windward climb time is 54.46% in one cycle. The average energy harvesting is 38.30%. For the leeward descent part, the average time is 22.82%, and energy harvesting is 61.70%. In the energy loss phases, the lower altitude turn phase takes an average time of 5.53% in one cycle and consumes 76.35% of energy. High altitude turn takes a long time but consumes less energy.

By summarizing the above results, it can be found that the UAV mass directly influences energy harvesting, and the maximum energy gaining increases linearly with the UAV mass increasing under the same flight mission. At the same time, there is little change in the trajectory, and the flight distance increases slightly in the lower altitude turn. For the dynamic soaring cycle, the mass increases, the energy harvesting during the windward climb and leeward descent increases, and the total energy increases. Specifically, in the dynamic soaring phases, it can be found that the windward climb and leeward descent are the energy-gaining phases. The windward climb takes the longest time and the leeward descent gains the most energy. High altitude turn and lower altitude turn are energy-consuming phases. High altitude turn phase takes longer than the lower altitude turn but consumes less energy.

It can be seen that for the dynamic gliding UAV, within the allowable range, the greater the mass, the more conducive to dynamic soaring. For the UAV with the planned dynamic soaring trajectory, the payload change has little influence on the track. During the flight, UAV should pay attention to the energy consumption in the lower altitude turn phase and use the windward climb to prolong the flight time while using leeward descent to accumulate the total energy of the UAV.

### 3.3. Influence of Initial Airspeed on Maximum Energy Harvesting

This section explored the influence of initial velocity change on the maximum energy gain of the UAV. The previous initial airspeed is 20 m/s, and the stall speed of the glider Fox is 7 m/s. Hence this paper selected the initial airspeed from 15 m/s to 25 m/s to compare the energy change. Figures 12 and 13 show the results of the energy variation and maximum energy gain achieved.

From Figures 12 and 13, the result shows that the initial airspeed increases, the flight time under the same flight mission increases, and the overall energy level of the UAV increases. Unlike the influence of mass, the relationship between initial airspeed and

maximum energy harvesting is no longer linear but presents a trend of increasing first and then decreasing. The maximum energy-gaining initial airspeed is about 23 m/s.

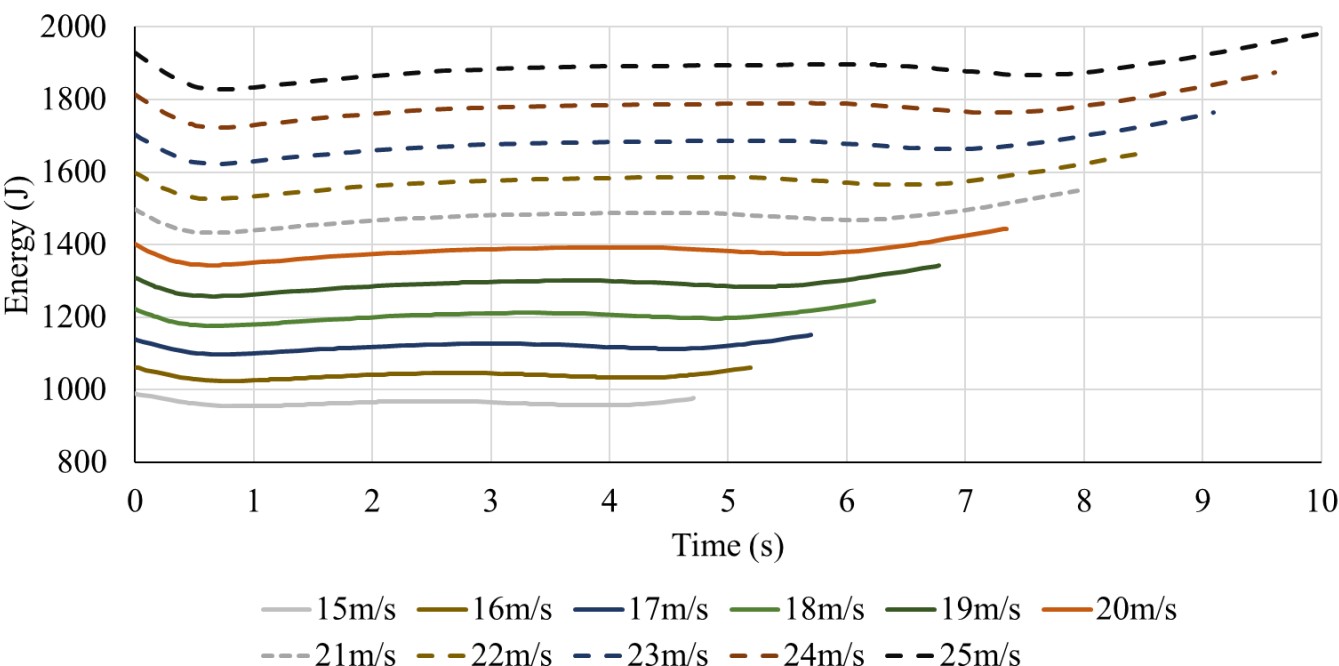

**Figure 12.** The energy variation curves in different initial airspeeds.

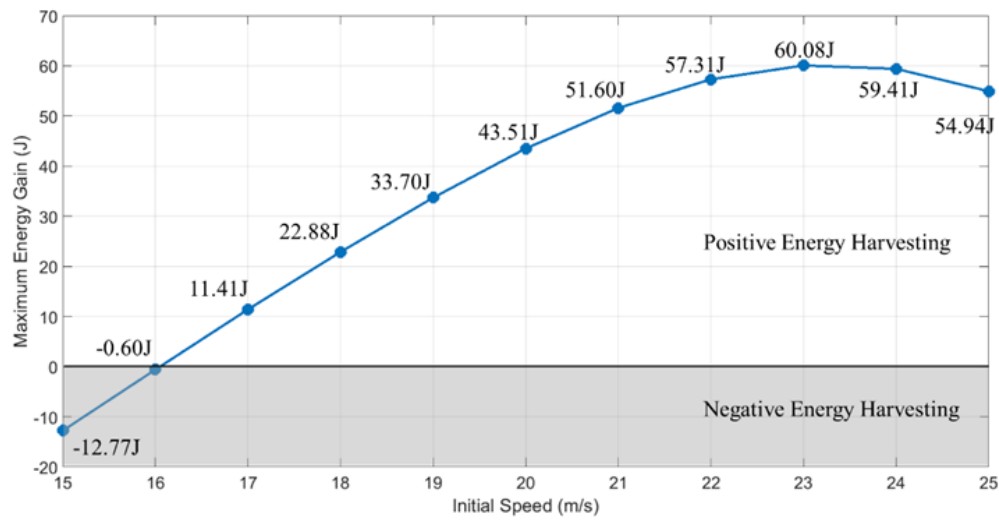

**Figure 13.** Total energy changes in different initial airspeeds.

Through the analysis of the energy harvesting mechanism model, the energy change of UAV is the integration of the cubic function of airspeed. Further, the energy change of UAV is the quartic with airspeed. From the function characteristics, it can be found that there is an optimal interval between the airspeed and maximum energy gain.

Next, observe the dynamic soaring trajectories in different initial airspeeds.

By observing the trajectories in Figure 14, it can be found that with the increase in initial airspeed, the UAV's dynamic soaring track is larger, and its coverage is broader. Compared with the 15m/s initial speed, the windward climb distance at 25 m/s increases by 281.73%, the crosswind flight distance at lower altitude turn increases by 20.49%, and the maximum altitude in flight also increases by 102.07%. Hence, the higher the initial airspeed, the longer the flight distance and time under the same task.

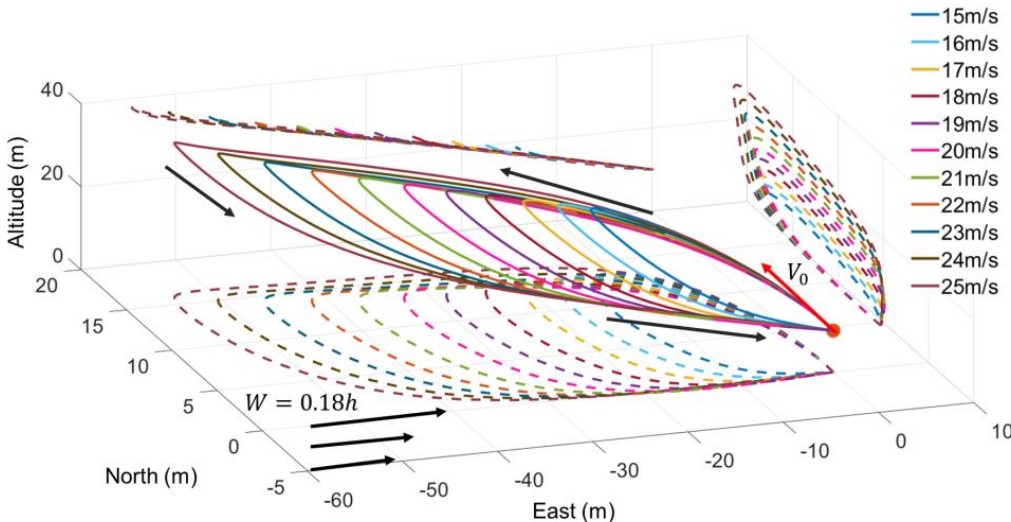

**Figure 14.** Trajectories for different initial airspeeds.

From the point of the four dynamic soaring phases, it is found that the initial airspeed increases, and the UAV's speed increases after the lower altitude turn, which means the speed at the beginning of the windward climb increases. After the windward climb, the UAV's airspeed is 8 m/s, which is close to stalling speed. Hence, the higher the initial airspeed, the greater the kinetic energy before climbing, the more energy can be consumed, and the time and distance both increase. Therefore, as the initial airspeed increases, the windward climb distance increases.

Based on the trajectory and total energy harvesting trend, energy changes in the four stages of dynamic soaring are observed in Figure 15.

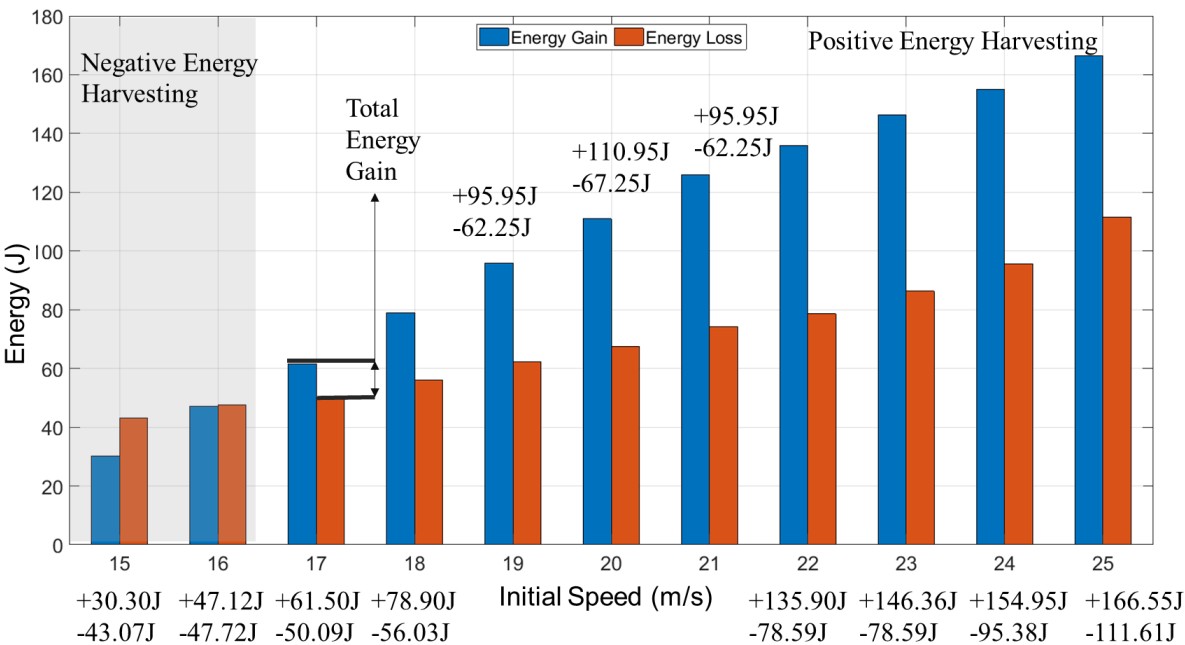

**Figure 15.** Comparison of energy gain and loss in different initial airspeeds.

The energy harvesting increased by 449.68%, and the energy consumption increased by 159.16% compared between 15 m/s and 25 m/s. Meanwhile, two initial airspeed character values appear, which are 16 m/s and 23 m/s: when the initial airspeed is under 16 m/s, the UAV cannot achieve enough energy from dynamic soaring. When the initial airspeed is 23 m/s, the UAV gains the maximum energy from dynamic soaring. Though the trend of

the energy changes, with the initial airspeed increase, the energy gain rate rises first and then decreases. The change rate of energy-consuming is lower than gaining but increases after 23 m/s.

Figure 16 shows the time of each phase, the results show that with the increase in initial speed, the time of windward climb and leeward descent increases, and the proportion of time in the whole dynamic soaring cycle increases. Both the time and proportion of lower altitude turn and high altitude turn decrease. These results can be analyzed from the increased airspeed in the two stages and the attitude changes during the turning time. In this paper, the UAV steering mainly depends on the rolling attitude change and lift (provides steering force). Meanwhile, the lift is closely related to airspeed. The higher speed leads to a greater steering force and a shorter time.

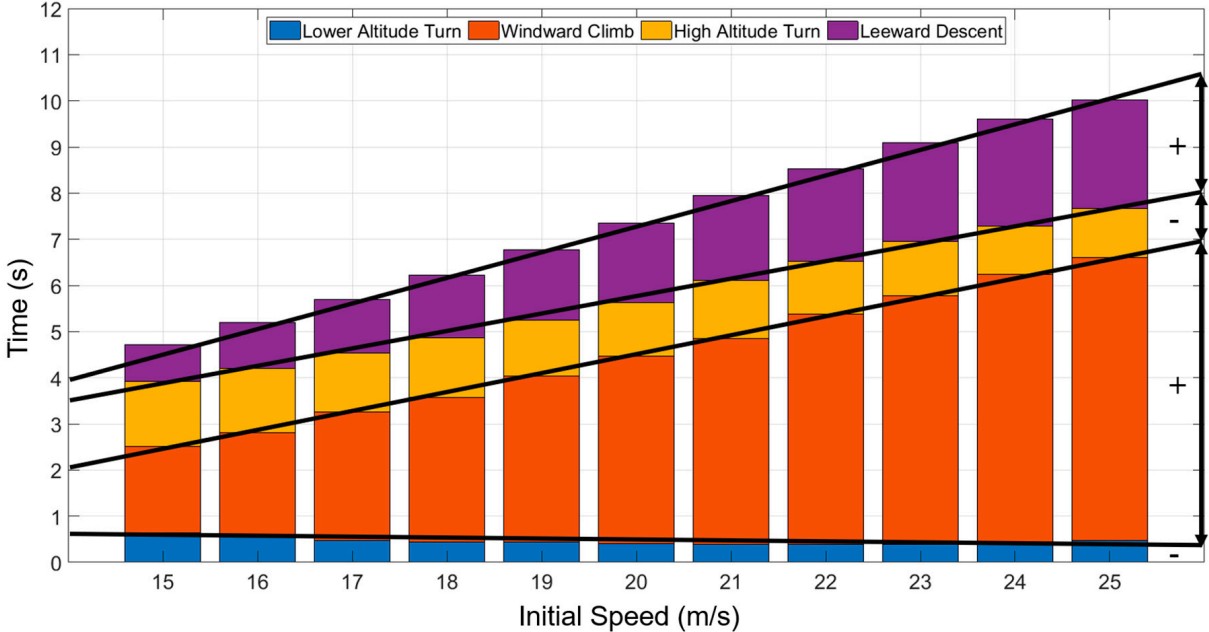

**Figure 16.** Time distribution in different initial airspeeds.

In conclusion, it can be found that the higher the airspeed of the UAV starts dynamic soaring, the more energy is obtained under the same flight mission. For this model, the UAV cannot achieve the flight mission without consuming its energy before the initial speed of 16 m/s, reaching the maximum energy gaining at 23 m/s. As for the flight trajectory, with the increase in initial speed, the flight distance and time of the UAV increases. Specific to each phase of dynamic soaring, it can be found that with the increase in initial airspeed, the energy harvesting rate in the windward climb and leeward descent first increases and then decreases, which is also reflected in the total energy change. Meanwhile, the time proportion of energy-gaining phases rises, and the proportion of consumption phases decreases.

Therefore, for the UAV applying dynamic soaring, determining the initial airspeed at the beginning can affect the quality of energy harvesting, and determining an optimal initial airspeed is conducive to the best energy harvesting of the UAV. Similarly, the increase in the initial airspeed positively impacts the flight distance and endurance.

### 3.4. Influence of Entering Angle on Maximum Energy Harvesting

The definition of the entering angle in this paper is the angle between the glider Fox's airspeed and the wind direction at the beginning of the dynamic soaring. This section studies the influence of different entering angles. In the above study, the wind field is from west to east (west wind), and the entering angle in these models is 90 deg. By changing the direction of the wind, the entering angles of −90 deg (270 deg), 0 deg, and 180 deg are

further explored with other UAV states unchanged. After calculation, the energy changes of the four different entering angles are shown in Figure 17 and Table 4.

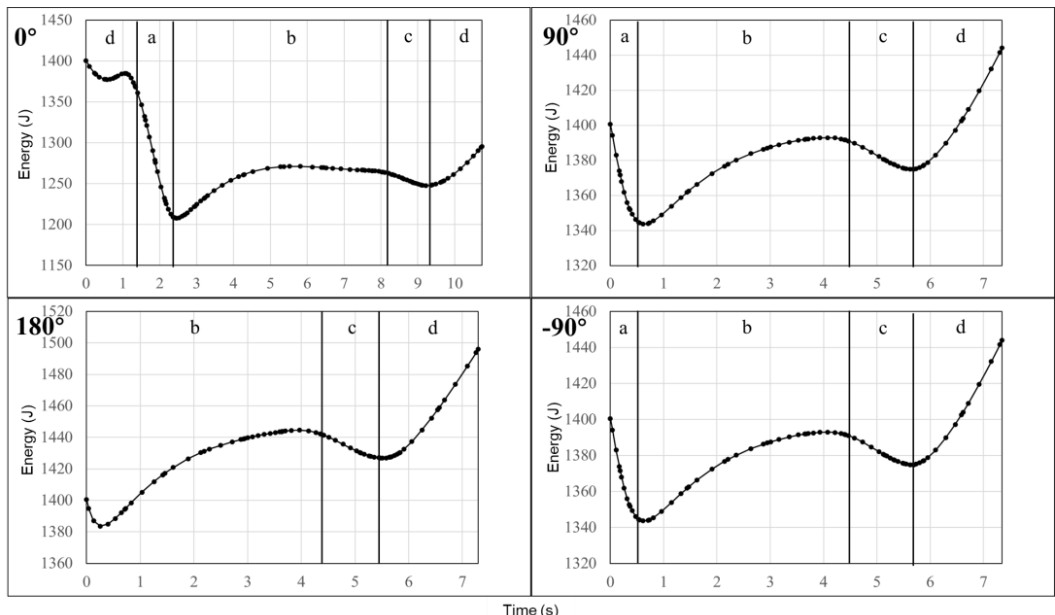

**Figure 17.** The energy variation curves in different entering angles as a function of time. (**a**) Lower altitude turn, (**b**) Windward climb, (**c**) High altitude turn, (**d**) Leeward Descent.

**Table 4.** Total energy changes in different entering angles.

| Entering Angle, Degree | Total Energy Changes, J |
|---|---|
| 0 | −133.77 |
| 90 | 43.51 |
| 180 | 95.50 |
| −90 | 43.51 |

The results show that when the entering angle is 180 deg, the UAV has the best energy harvesting through dynamic soaring, while when the entering angle is 0 deg, it is the worst. The energy gained in the 90 deg and −90 deg is the same. Further, the trajectories of four entering angles are plotted in Figure 18.

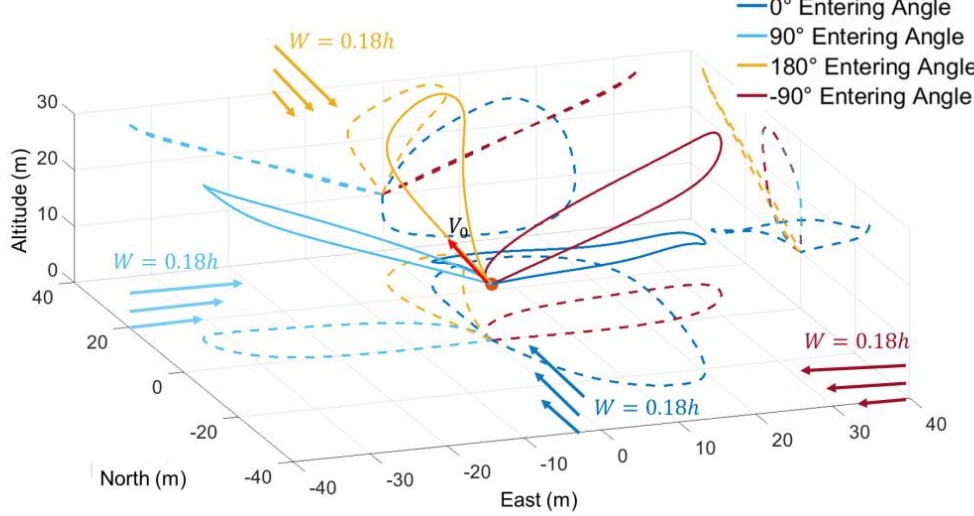

**Figure 18.** Trajectories for different entering angles.

From the trajectory perspective in Figure 18, when the entering angle is 180 deg, the UAV starts at the windward climb. Then the UAV completes the high altitude turn and leeward descent. The whole dynamic soaring cycle is missing the lower altitude turn phase. When the entering angle is ±90 deg, the UAV starts at the lower altitude turn and begins the windward climb swiftly. Only when the entering angle is 0 deg, the UAV starts at the leeward descent. Further, the trajectories are axisymmetric along the north axis at the entering angles of ±90 deg.

From the time perspective, compared with other phases, the windward climb takes the largest proportion in all different entering angles shown in Figure 19. Meanwhile, the 180 deg entering angle (best energy harvesting) has the biggest windward climb time proportion (58.52%) of the other three angles. Figure 20 shows that no matter what the entering angle is, the energy gain in the windward climb phase remains unchanged. The entering angle influences the energy change in the lower altitude turn phase. The 0 deg entering angle has the longest lower altitude turn time and energy consumption, while the 180 deg has no turning phase.

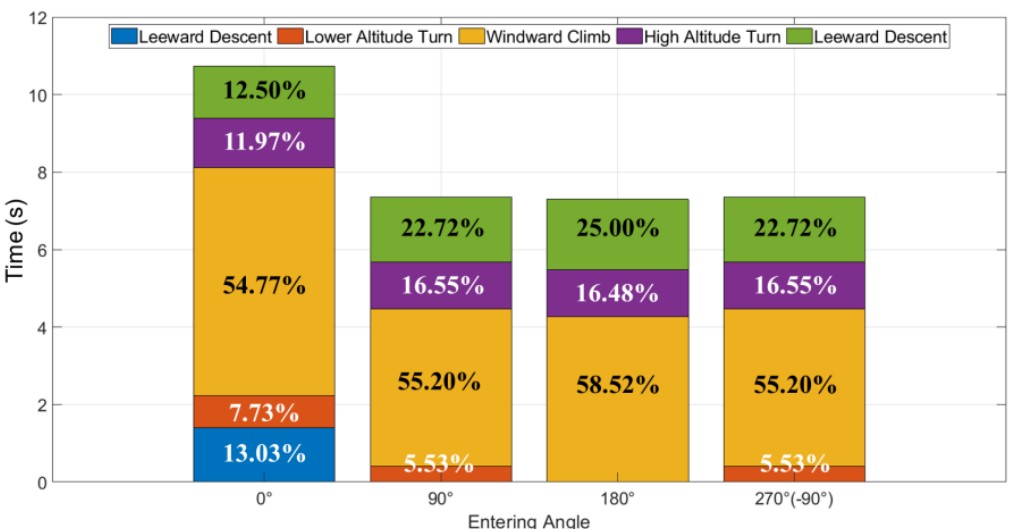

**Figure 19.** Time distribution in different entering angles.

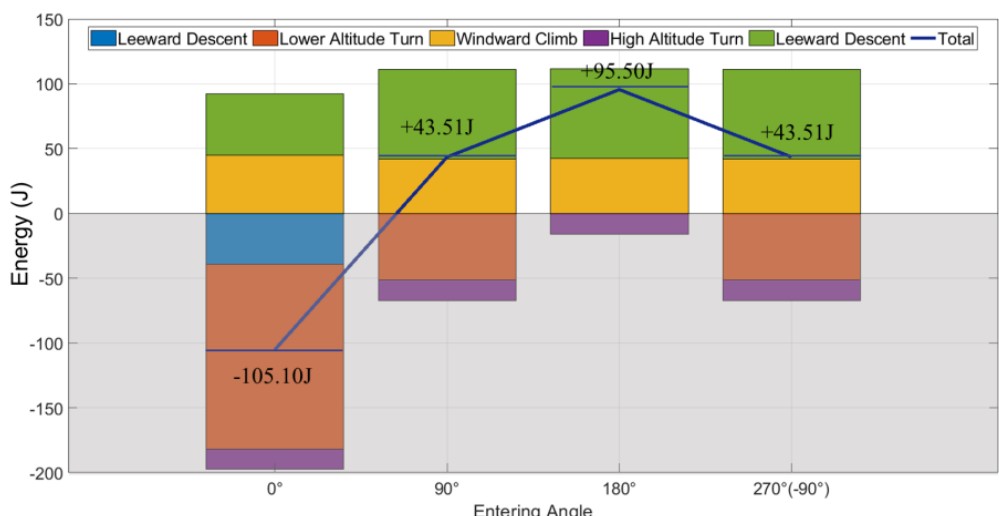

**Figure 20.** Energy change distribution in different entering angles.

Hence summarizing the results, it can be found that the UAV entering angle influences the dynamic soaring energy harvesting. For this study, the maximum energy gains at 180 deg entering angle, and the minimum at 0 deg. The angles of ±90 deg have the same

trend both in the energy harvesting and the trajectory. For the trend of the dynamic soaring cycle, the UAV will take advantage of the energy harvesting as soon as the windward climb starts. This trend can be verified in the ±90 deg entering angles: when the UAV is under a crosswind situation, it will turn to the windward climb phase rather than the leeward descent phase. The counts of the phases are also different in the different entering angles. The total phase number in the 0 deg is 5 while the 180 deg is 3 (missing the lower altitude turn), much less than the 0 deg condition. As for the time ratio and energy change in the dynamic soaring cycle, the windward climb phase still has the longest time proportion. The energy gain or loss is not changed in the windward climb phase and high altitude turn phase at different entering angles, while the other two phases change clearly.

For the dynamic soaring application, it should try to choose windward or crosswind at the beginning of the dynamic soaring and adjust its attitude as soon as possible to enter the windward climb phase to perform better energy harvesting. Meanwhile, it should avoid downwind entering the wind field and unnecessary lower altitude turn, thus reducing the energy loss.

### 3.5. Discussion for UAV and Flight Strategy Design

This paper shows multiple parameters special in dynamic soaring. Results illustrate that UAV weight is vital for dynamic soaring energy harvesting. The interesting part is that weight is usually a negative influence in UAV design. And the weight of the battery carried by a long-endurance UAV is more than 50% of the weight of the whole aircraft. Hence the positive influence of weight on dynamic soaring is an advantage to allowing UAVs to carry more payloads.

For dynamic soaring UAV design, the influence of weight also relaxed the weight constraint. The designer needs to balance energy harvesting efficiency and flight performance concerning weight. Meanwhile, UAVs' weight change does not obviously change dynamic soaring trajectories. This means during the flight, UAV does not need to change flight strategy when payloads drop.

Meanwhile, the above results and rules give a new idea for dynamic soaring flight strategies. Some of the most favorable parameters for energy harvesting emerge, such as optimal initial airspeed. These parameters in a UAV will not change if the flight mission is determined. Hence, before designing a new flight strategy for a UAV, first defining these special values will help improve the strategy's feasibility. On the one hand, these values can help to determine whether the dynamic soaring can be realized according to the current environment and mission. On the other hand, combining these parameters in flight strategy, such as set range for airspeed, can get a more reliable flight strategy. Further, previous studies which were based on the flight parameters to guide UAVs can use these special parameters to improve control effects.

The influence regulation also inspires a repetitive flight strategy. The initial airspeed only affects the range of the trajectory but not the shape. After using trajectory optimization to get a dynamic soaring trajectory, it can guide UAV flight with different initial airspeeds. Further, combined with the character values and basic trajectory shape, the UAV can determine a new track quickly. These will help to improve solution time for real-time trajectory optimization.

Detailed analysis also inspires dynamic soaring flight strategy. The windward climb phase takes the longest time, and the leeward descent phase gains the most energy. Hence, a flight strategy design needs to increase the number of these two phases. Moreover, UAVs need to enter the windward climb as soon as possible at the beginning. The perfect initial state is windward at the optimal airspeed. Meanwhile, the flight strategy needs to reduce the time of turning phases, especially the lower altitude turn, to obtain more energy.

### 4. Conclusions

The UAVs' parameters significantly influence dynamic soaring performance, the regularity is vital for dynamic soaring UAV design and flight strategies plan. This paper

selected UAV mass, initial airspeed, and entering angle as major influence factors based on the dynamic soaring energy harvesting mechanism model. By using trajectory optimization, the trajectory and flight parameters variation were obtained. This paper analyzed these results and summarized available rules to improve dynamic soaring flight strategies. The main findings of this paper are as follows:

1. The effect of UAV mass change mainly shows on the energy side. With the change in UAV's weight, the whole energy level is rising. Meanwhile, the energy gained from dynamic soaring increases linearly, which matches the mechanism model. However, UAV mass change has no obvious impact on the soaring trajectory. In detail, the flight distance and each phase time are changed a little with the mass increasing. This result enlightens the benefits of dynamic soaring energy harvesting with greater mass. And during the flight, the payload change will not influence the planned trajectory;

2. The effect of a UAV's initial airspeed on dynamic soaring is significant. The initial airspeed is increased to 25 m/s from 15 m/s. The total energy level is rising, but the energy harvesting rule is different from mass change. The relationship between initial airspeed and maximum energy gained is a convex curve. When the initial airspeed is below 16 m/s, the UAV cannot gain energy from dynamic soaring. While the initial airspeed is 23 m/s, the UAV reaches maximum energy harvesting. The dynamic soaring trajectory also has a noticeable change while initial airspeed increases. The average flight distance increases by 134.76% in 25 m/s initial airspeed compared with the 15 m/s. Moreover, the endurance also increases by 114.89%. In detail, the windward climb phase is a major growth item. The result shows that UAVs have specific initial airspeed to determine whether energy harvesting can be achieved. Although the energy level increases when the initial airspeed rises, there is an optimal initial airspeed to achieve maximum energy gain. With the initial airspeed increases, the major change in detail soaring phases is the windward climb phase. Therefore, determining the optimal initial airspeed at the beginning is vital for UAV dynamic soaring flight strategy and conducive to the best energy harvesting. Meanwhile, the increase in the initial airspeed positively impacts the flight distance and endurance;

3. The effect of a UAV's entering angle on dynamic soaring is significant. The result shows that 180 deg entering angle is best for dynamic soaring energy harvesting and 0 deg is the worst. An interesting part is that 90 deg and −90 deg have the same energy variation. The trajectory also has significant change while entering angle changes. The first phase is different in different entering angles. At 180 deg, the UAV starts at the windward climb phase and completes the high altitude turn and leeward descent phase. At ±90 deg, the UAV starts at the lower altitude turn phase and begins the windward climb swiftly. Only when the entering angle is 0 deg, the UAV starts at the leeward descent. The results show that the windward climb phase is vital for dynamic soaring and takes the largest proportion in all different entering angles;

4. The results enlighten the UAV dynamic soaring application from the design and strategy sides. UAV weight needs to be balanced between energy harvesting and flight performance. This gives a new design space for long-endurance UAV. The key parameters, such as optimal initial airspeed, can be obtained before the flight and used to improve the dynamic soaring strategy. This paper also shows that the windward climb and leeward descent phases are vital and need to be considered in flight strategy. In general, UAVs should increase the climb time, reduce the turning time, and enter the windward climb phase primary.

This paper simplified some conditions that need to be carried out in the future. In this paper, the entering angles are considered horizontal and ignored the vertical angle. Meanwhile, this paper only considered direct crosswind, windward, and leeward, which cause large intervals between different entering angles. These blank needs to be reduced for a more precise entering angles study. Moreover, the energy harvesting mechanism model includes many parameters. This paper only selected three of them, but others also have influence characteristics of dynamic soaring performance. Furthermore, the regularity

of the parameters that can change during the flight may contribute more to the dynamic soaring flight strategies.

**Author Contributions:** Conceptualization, W.W.: methodology, software, writing—original draft preparation, W.A.: writing—review and editing, B.S.: writing—review and editing, supervision. All authors have read and agreed to the published version of the manuscript.

**Funding:** This research received no external funding.

**Institutional Review Board Statement:** Not applicable.

**Informed Consent Statement:** Not applicable.

**Data Availability Statement:** Data available in a publicly accessible repository.

**Conflicts of Interest:** The authors declare no conflict of interest. The funders had no role in the design of the study; in the collection, analyses, or interpretation of data; in the writing of the manuscript, or in the decision to publish the results.

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
