# Peer review of "Dynamic Soaring Parameters Influence Regularity Analysis on UAV and Soaring Strategy Design"

_drones, doi:10.3390/drones7040271_

Round 1

Reviewer 1 Report

The paper shows a study of different parameters influence on dynamic soaring of long range UAV. Improvements on the methodology description are required in such a way that the reader is provided with the possibility of understanding and replicating the processes shown. All figures must be referenced and linked in the text some of them are missed ex. Figure 1. 

Author Response

Thank you to point out. I have rewritten my sentence to be more clear about the figures' reference. 

I have rewritten section 2 with more method information.

Reviewer 2 Report

This paper selected UAV mass, initial airspeed, and entering angle as major influence factors based on the dynamic soaring energy harvesting mechanism model. Wish in the future, the other factors those also influence characteristics of dynamic soaring performance can be studied by the authors.

  1. Can the dynamic soaring flight help an UAV reach the altitude of pseudo-satellites?
  2. There is a type error in Eq. (5).

Author Response

Thank you to point out,I have corrected error you mentioned.  In all dynamic soaring can help UAV reach the altitude of pseudo-satellites but needs additional power to achieve better performance. Further disscussion please see the attachment.

Reviewer 3 Report

This manuscript proposes an analysis of the influence of key parameters on the efficiency of the dynamic soaring, in terms of harvested energy in each cycle. While the results are interesting, details on modeling and methods are need improvements. This article has the potential to be published after the proposed changes below.

1- It is not clear how the aerodynamic forces for the specific UAV are included in the energy harvesting equation, there is a lack of data about the Fox glider lift and drag coefficients as function of angle of attack. 

2- When changing UAV mass, is the effect on the drag coefficient computed?

3- Check the caption of figure 3, it does not seem to adequately describe the 3-view drawing of the UAV.

Author Response

Thanks for your comments, 3-view drawing figure is corrected. Two major problems are discussed in the attachment, please see the attachment. Thank you again!

Reviewer 4 Report

See file attached below.

Author Response

Thank you for your feedback. Thanks to point out the errors. I have corrected these errors. I also remain some problems unchanged, please see the attachment.
